# Correlation between Three-Dimensional Volume and Malignant Potential of Gastrointestinal Stromal Tumors (GISTs)

**DOI:** 10.3390/jcm9092763

**Published:** 2020-08-26

**Authors:** Jin Hwa Park, Bo-kyeong Kang, Hang Lak Lee, Jai Hoon Yoon, Kang Nyeong Lee, Dae Won Jun, Oh Young Lee, Dong Soo Han, Byung Chul Yoon, Ho Soon Choi

**Affiliations:** 1Department of Internal Medicine, Asan Medical Center, Seoul 05505, Korea; pjh6718@hanmail.net; 2Department of Radiology, Hanyang University School of Medicine, Seoul 04763, Korea; dr.bokyeong.kang@gmail.com; 3Department of Internal Medicine, Hanyang University School of Medicine, Seoul 04763, Korea; jaihoonyoon@hanyang.ac.kr (J.H.Y.); leekn@hanyang.ac.kr (K.N.L.); noshin@hanyang.ac.kr (D.W.J.); leeoy@hanyang.ac.kr (O.Y.L.); yoonbc@hanyang.ac.kr (B.C.Y.); hschoi96@hanyang.ac.kr (H.S.C.); 4Department of Internal Medicine, Hanyang University School of Medicine, Guri 11923, Korea; hands@hanyang.ac.kr

**Keywords:** GIST (gastrointestinal stromal tumors), malignancy, volume, size

## Abstract

Background and purpose: Gastrointestinal stromal tumors (GISTs) are rare diseases of the gastrointestinal tract but they are the most common gastrointestinal tumors of mesenchymal origin. Since most GISTs have malignant potential, their probability of malignant progression must be evaluated. This study was conducted to examine the correlation between three-dimensional GIST volume measured by CT and malignant potential. Materials and methods: A retrospective study was performed on 70 patients diagnosed with GIST after surgical resection in Hanyang University Seoul Hospital from 2012 to 2017. Linear regression analysis was used to establish which between the length of the long axis of GISTs, originally considered a predictor of malignancy, and their volume was a more accurate predictor of malignancy. Tumor dimensions were measured by CT. Results: Data were analyzed using the chi-square test or Student’s *t*-test and logistic regression. Of the GISTs, 53 (75.71%) were in the stomach, 3 (4.29%) in the small intestine, and 14 (20.0%) in the large intestine. The mean age of the malignant GIST group was significantly higher than that of the benign GIST group (*p* = 0.032), their tumor long axes were significantly greater (*p* = 0.073), their tumor volumes were significantly larger (*p* = 0.001), and the frequency of tumor necrosis was higher (*p* = 0.001). In multivariate analysis, malignant GIST was associated with location in organs other than the stomach (OR 7.846, 95% CI 1.293–47.624, *p* = 0.025), longer axis (OR 1.037, 95% CI 1.011–1.065, *p* = 0.006), larger volume (OR 1.003, 95% CI 1.000–1.006, *p* = 0.029), and necrosis (OR 12.222, 95% CI 1.945–76.794, *p* = 0.008). The mean age of the recurrent GIST group was higher than that of the non-recurrent group (*p* = 0.045), their tumor long axes were significantly longer (*p* = 0.005), and their volumes were greater, but this last difference was not significant (*p* = 0.072). Conclusions: Tumor volume can be considered an additional risk factor in assessing the malignant potential of GISTs and tends to increase in recurrent GISTs.

## 1. Introduction

Gastrointestinal stromal tumors (GISTs) are rare diseases of the gastrointestinal tract but they are the most common digestive tract tumors of mesenchymal origin, accounting for 50% of submucosal tumors [1]. According to previous studies conducted in Iceland and Sweden, the incidence of GIST is 11–14.5 per 100,000 subjects per year [2]. With the recent advances in diagnostic techniques and the accuracy of diagnosis, the number of patients being diagnosed is increasing [3]. Since GISTs have malignant potential and can recur, it is important to evaluate them at the time of diagnosis. In 2002, the National Cancer Institute (NIH) published a consensus on GIST recurrence risk. They proposed anatomic site, size, mitotic rate, rupture, and location as conventional risk factors for GIST and divided recurrence potential into four risk groups according to the presence of these risk factors [3,4]. In addition, as studies have recently been published that necrosis is related to malignant potential, necrosis is also suggested as an important risk factor [5,6].

Recent technical advances have suggested that it is helpful to evaluate three-dimensional tumor burden by PET and MRI [7]. Since GISTs are not spherical, it was thought that assessing tumor volume would be more helpful than measuring tumor size (longest dimension). Therefore, recent studies have used volume for evaluating the metastatic and malignant potential of GISTs [8,9,10]. The Choi criteria have been developed for assessing responses to imatinib in GIST patients, and a method of evaluating responses by combining morphological tumor response and biological response (tumor density, CT attenuation coefficient (Hounsfield units [HU]) has been presented. The theoretical assumption on which these criteria are based is that volume is more sensitive than a one-dimensional measurement because it has a wider dynamic range [11,12].

Although it has been suggested that tumor volume can be used as a method for GIST evaluation, this issue has not been much studied. It is difficult to use previous data to test this hypothesis because of limitations such as that only patients with metastasis were examined, tumor diameters were not evaluated by experts, and follow-up periods were short. The aim of this study was to determine whether tumor three-dimensional volume is helpful in predicting the malignant potential of GISTs.

## 2. Materials and Methods

### 2.1. Study Design

This study was performed on adult patients over 18 years of ages who underwent surgical or endoscopic resection and were diagnosed with GIST in Hanyang University Medical Center from 2012 to 2017. The study was approved by the Hanyang University Review Committee. (HYUH 2017-07-045-005) and is a retrospective study of electronic records in the Hanyang University Medical Center from 1 January 2012 to 31 December 2017.

### 2.2. Data Source

The longest axis, traditionally used as a malignancy predictor, and three-dimensional volume, the potential predictor examined in this study, were measured by CT performed by a specialist in radiology. Through the cooperation of the radiology department, the patient’s CT was reconstructed into 3D, and the radiology specialist evaluated the 3D volume through the reconstructed CT.

Baseline characteristics such as patients’ sex, age, and location of GIST based on endoscopic images and descriptive results were investigated. Patients who underwent EUS (Endoscopic ultrasonography) were evaluated for pattern, density, induced layers, subtype, and mitotic counts (per 50 HPF (High power field)) examined by histology.

### 2.3. Outcomes

Primary outcome was defined as malignant change, and secondary outcome was defined as recurrence. Malignant changes at the time of diagnosis, duration of follow-up, and recurrence during follow-up were evaluated. Previous studies have defined tumor necrosis as the presence of microscopic coagulation necrosis without inflammation or fibrosis, regardless of the proportion of tumor necrosis to tumor cells. Tumor necrosis was defined by referring to previous studies.

### 2.4. Statistical Analysis

We used Microsoft Office Excel 2010 (Microsoft, Redmond, WA, USA) and IBM SPSS Statistics version 25.0 (IBM, New York, NY, USA) for data processing. Data are expressed as median values, and differences in variables between the pairs of groups were analyzed using the chi-square test or Student’s *t*-test. To examine factors associated with GIST malignancy potential, we used logistic regression and analyzed Hosmer–Lemeshow chi-square values. Odds ratio and 95% confidence intervals were calculated for each variable. All reported *p* values were 2-tailed, and the significance level was set at 0.05.

## 3. Results

### 3.1. Clinical Characteristics of Patients with GISTs

A total of 70 patients, aged 18 years or older, who underwent surgical or endoscopic resection in Hanyang University Medical Center from 2012 to 2017 were diagnosed with GIST. Six of the GISTs (8.57%) were malignant at the time of diagnosis. Patients with malignancy at the time of diagnosis were significantly older than those without malignancy (60.83 ± 20.57 vs. 57.77 ± 11.79, *p* = 0.032), and the malignancies were more frequent in non-stomach organs than in the stomach (*p* = 0.011). The long axes of tumors were significantly greater in the patients with malignancies (73.33 ± 38.04 vs 35.40 ± 22.38, *p* = 0.073), tumor volumes were also larger (270.67 ± 410.80 vs 53.07 ± 142.84, *p* = 0.001), and the frequency of necrosis was higher (66.7% vs 14.1%, *p* = 0.001). There was no significant difference in gender or mitotic count (<5 versus ≥5) (Table 1).

### 3.2. Clinical Characteristics of Patients with Recurrent GISTs

Twenty-two patients were not followed up, and four of the patients that were followed up (8.33%) suffered recurrence. Mean follow-up was 3.79 years, and there was no statistically significant difference in the follow-up period between the patients who suffered recurrence and those who did not. Patients in the former group were older than those in the non-recurrent group (61.00 ± 19.55 vs. 58.41 ± 10.81, *p* = 0.045), and recurrence was more common in non-stomach organs than in the stomach (*p* < 0.001). In the group with recurrence, the long axes of the original tumors were longer than in the group without recurrence (72.49 ± 22.01 vs. 33.67 ± 25.08, *p* = 0.005), and their volumes were larger, though this difference was not significant (240.96 ± 242.38 vs. 56.25 ± 188.37, *p* = 0.072). Gender and mitotic count were not significantly different (Table 2).

### 3.3. Multivariate Analysis of the Malignant Potential of GISTs

In a multivariate analysis (Table 3), malignant tumors were more common in organs other than the stomach (OR 7.846, 95% CI 1.293–47.624, *p* = 0.025) and had longer axes (OR 1.037, 95% CI 1.011–1.065, *p* = 0.006). Their volumes were also larger (OR 1.003, 95% CI 1.000–1.006, *p* = 0.029). When the tumors were divided into two groups, one comprising those with long axes ≥5 cm and the other those with long axes <5 cm, it was found that the frequency of malignancy was significantly higher in the former (OR 11.778, 95% CI 1.873–74.05, *p* = 0.009). The incidence of necrosis was higher in malignant tumors (OR 12.222, 95% CI 1.945–76.794, *p* = 0.008).

### 3.4. Receiver Operating Characteristic (ROC) Curve of the Malignant Potential of GISTs

The tumor long axis and tumor volume were compared for their potential as malignancy predictors, and ROC curves were drawn to measure area-under-the-curve (AUC) values. The tumor long axis was found to have an AUC of 0.876 (95% CI 0.771–0.982), and the tumor volume was found to have an AUC of 0.89 (95% CI 0.796–0.984). These values were not significantly different, but the positive predictive value of the tumor volume was slightly higher than that of the tumor long axis (Figure 1).

## 4. Discussion

In this study, it was shown that tumor volume can be used as a risk factor for GIST malignancy and tends to increase in recurrent GISTs.

About 50% of submucosal tumors are GISTs. Since the need for additional treatments, such as the administration of imatinib and simple resection, is dependent on the risk of malignancy or recurrence, it is important to assess this risk at the time of tumor diagnosis [2]. CT, MRI, and PET–CT have been used in the diagnosis of diseases such as GIST, and radiological examination can be helpful in evaluating risk and therapeutic responses. Tumor size and mitotic count are conventional risk factors [3,4]. However, recent advances in diagnostic techniques allow more detailed 3D image evaluations, and CT, MRI, and PET–CT are thought to be helpful in evaluating the risk and treatment responses for tumors such as GISTs that lack spherical features.

Several studies have been conducted on risk factors for GIST malignancy and recurrence. In 2002, a method of classification (very low, low, intermediate, high risk) was proposed, based on tumor size and mitotic count, and has been used up to the present [4]. According to a study published by Dematteo et al. in 2008, a high recurrence rate is correlated with more than five mitoses per high-power field, tumor length ≥10 cm, and tumor location [13]. In a study conducted by Jumniensuk, metastasis was more frequent in tumors ≥10 cm (*p* = 0.023), tumors composed of non-spindle cells (*p* = 0.027), tumors with mitotic count ≥5 (*p* = 0.000), and tumors with myxoid change (*p* = 0.002); recurrence was associated with myxoid change (*p* = 0.045) [2]. Recently, studies including meta-analyses have been published, indicating that tumor necrosis is a risk factor for malignant GIST and is also thought to play an important role in malignancy [5,6]. In this study, necrosis was also identified as an independent risk factor for malignant GIST.

Recently, studies evaluating risk have been carried out using tumor volumes calculated from 3D images. A study of 56 patients with liver metastasis suggested that 3D volume was a better method to evaluate the risk of liver metastasis after imatinib treatment than the tumor long axis [14]. In addition, a recent study of patients with metastatic as well as non-metastatic GIST found that tumor volume was a better measure of the risk of recurrence than the maximum length or the mitotic count [9].

A previous study evaluating the degree of predictability of tumor volume was limited by the small number of patients and by the fact that diameter evaluation was not carried out by experts. Our study shows that tumor volume is an effective measure of the risk of malignancy in patients with GIST. Three-dimensional reconstruction and volume measurement are possible in standard centers. Therefore, GIST volume measurement can be a practically useful method to establish treatment strategies.

However, this study has several limitations. It was performed in a single institution, so the number of patients was quite small; although the usefulness of tumor volume measurements was established, the accuracy of the appropriate cut-off value was limited, as was the accuracy of the measurement of risk factors associated with recurrence. Further studies with a large number of patients are needed to fully evaluate risk factors associated with recurrence.

In conclusion, measuring tumor volume at the time of diagnosis in GIST patients may be helpful to predict malignancy and recurrence.

## Figures and Tables

**Figure 1 jcm-09-02763-f001:**
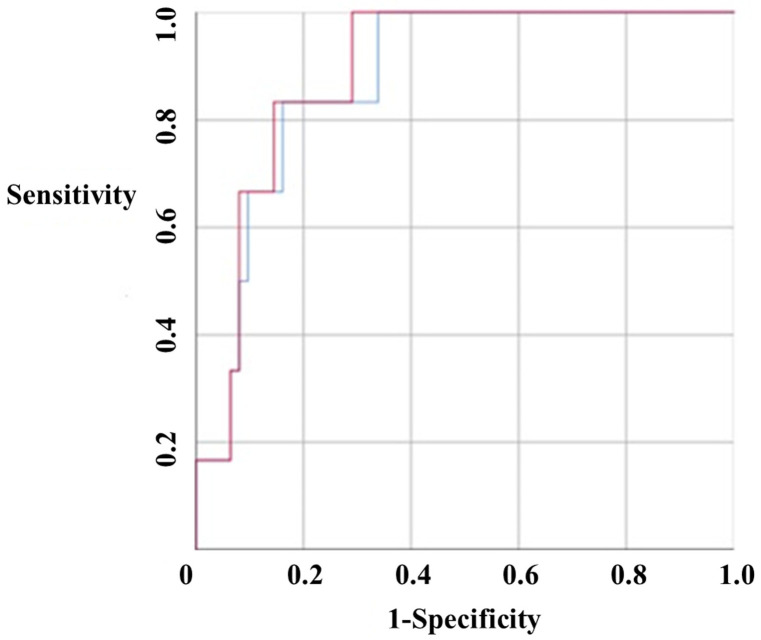
Receiver operating characteristic (ROC) curve. The tumor long axis was found to have an area under the curve (AUC) of 0.876 (red color, 95% CI 0.771–0.982), and the volume was found to have an AUC of 0.89 (blue color, 95% CI 0.796–0.984). These values were not significantly different, but the positive predictive value of tumor volume was slightly higher than that of tumor long axis.

**Table 1 jcm-09-02763-t001:** Clinical characteristics of patients with GISTs.

	Benign GIST (*n* = 64)	Malignant GIST (*n* = 6)	*p*
Age	57.77 ± 11.79	60.83 ± 20.57	0.032
Sex			
Men	30 (46.9%)	4 (66.7%)	0.354
Women	34 (53.1%)	2 (33.3%)	
Location			
Stomach	51 (80.0%)	2 (33.3%)	0.011
Non-stomach	13 (20.0%)	4 (66.7%)	
Mitoses			
Less than 5	52 (81.3%)	4 (66.7%)	0.393
5 or more	12 (18.7%)	2 (33.3%)	
Long axis (mm)	35.40 ± 22.38	73.33 ± 38.04	0.073
Volume (cm^3^)	53.07 ± 142.84	270.67 ± 410.80	0.001
Necrosis	9 (14.1%)	4 (66.7%)	0.001

Data are expressed as mean ± SD (No.%); *p*, *p*-value; GIST, gastrointestinal stromal tumor.

**Table 2 jcm-09-02763-t002:** Clinical characteristics of patients with recurrent GISTs.

	No Recurrence (*n* = 44)	Recurrence (*n* = 4)	*p*
Age	58.41 ± 10.81	61.00 ± 19.65	0.045
Sex			
Men	20 (45.5%)	3 (75.0%)	0.257
Women	24 (54.5%)	1 (25.0%)	
Follow-up period (year)	3.84 ± 2.73	3.75 ± 1.26	0.950
Location			
Stomach	36 (81.8%)	0 (0.0%)	0.000
Non-stomach	8 (18.2%)	4 (100.0%)	
Mitoses			
Less than 5	33 (75.0%)	3 (75.0%)	1.000
5 or more	11 (25.0%)	1 (25.0%)	
Long axis (mm)	33.67 ± 25.08	72.49 ± 22.01	0.005
Volume (cm^3^)	56.25 ± 188.37	240.96 ± 242.38	0.072

Data are expressed as mean ± SD (No. %).

**Table 3 jcm-09-02763-t003:** Multivariate analysis of factors related to malignancy.

Variable	Odd Ratio (95% CI)	*p*
Age	1.020 (0.953–1.092)	0.567
Location		
Stomach	Reference	0.025
Non-stomach	7.846 (1.293–47.624)	
Long axis	1.037 (1.011–1.065)	0.006
Volume	1.003 (1.000–1.006)	0.029
Long axis		
Less than 5cm	Reference	0.009
Above 5cm	11.778 (1.873–74.05)	
Necrosis	12.222 (1.945–76.794)	0.008

Data are expressed as *p*-value and 95% CI. CI, confidence interval.

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
