# Peer review of "Correlation between Three-Dimensional Volume and Malignant Potential of Gastrointestinal Stromal Tumors (GISTs)"

_jcm, 2020, doi:10.3390/jcm9092763_

Round 1
Reviewer 1 Report
In this study Park et al attempt to correlate predicted 3D volume of GIST tumors to evaluate malignant potential. In short, this study in somewhat limited by low patient numbers, but this is however a rare disease. Tumors with a long axis over 5 cm are slightly more likely to be malignant, consistent with the literature (https://www.ncbi.nlm.nih.gov/pmc/articles/PMC1420965/).
More promise rather than volume of tumor for malignant potential is volume of necrosis seen. This has been the subject of a recent meta-analysis (https://journals.lww.com/md-journal/FullText/2019/04260/Prognostic_value_of_tumor_necrosis_in.57.aspx).
To make this work more impactful, I would respectively ask that the authors calculate an estimate of the volume of radiographically determined necrosis volume is applicable tumor and use this within a multivariable analysis similar to their Table 3. If numbers are insufficient for such an analysis, then I would suggesting adding necrosis as a variable to tables 1 and 2.
Author Response
Thank you for your good comment. According to your opinion, we investigated by adding necrosis as a variable. Necrosis is one of the risk factor malignant GISTs and a number of studies were confirmed necrosis through pathological results. We asked the pathology department to additionally check the necrosis of each case and added necrosis as a variable. Related parts were mentioned from page 3, line 58 to page 3, line 60, from page 4, line 90 to page 4, line 93, page 5, line 110 & page 6, 134-135, from page 8, line 167 to page 8, line 170, and table 1 & 3.
… In addition, as studies have recently been published that necrosis is related to malignant potential, necrosis is also suggested as an important risk factor.
… Previous studies have defined tumor necrosis as the presence of microscopic coagulation necrosis without inflammation or fibrosis, regardless of the proportion of tumor necrosis to tumor cells. Tumor necrosis was defined by referring to previous studies.
… The long axes of tumors were significantly greater in the patients with malignancies (73.33 ± 38.04 vs 35.40 ± 22.38, p = 0.073), tumor volumes were higher (270.67 ± 410.80 vs 53.07 ± 142.84, p = 0.001), and frequency of necrosis was higher (66.7% vs 14.1%, p=0.001).
… The incidence of necrosis was higher in malignant tumor (OR 12.222, 95% CI 1.945-76.794, p=0.008).
… Recently, studies including meta-analysis have been published that tumor necrosis is a risk factor of malignant GIST, and necrosis is also thought to play an important role. In this study, necrosis was also identified as an independent risk factor of malignant GIST.
Reviewer 2 Report
Dear Authors,
Gastrointestinal stromal tumors are the most common pathologies among subepithelial lesions of GI tract with the mainly low risk of malignancy, especially with the gastric location; however, a couple of factors have been estimated till now to predict the malignant course.
This study touches on the new factor like the three-dimensional GIST volume related to clinical implication.
Therefore I would like to highlight a couple of fields.
Major
The study concept was well prepared; however, this study doesn't show a clear way of relation with malignancy. In the manuscript, two definitions are used- recurrence and malignancy, which were not clearly defined. The reader can be confused, especially when in Table no 1, benign and malignant lesions were presented.
Also, there is no visible what patients were included to follow up in the method. Therefore, I propose to change and organize the method part, by adding primary and secondary outcomes.
Minor
Based on the manuscript, the main tool, like the GIST volume, was not presented as a standardized measurement method? This main tool could be wider presented in the method/material part.
The recommendations for GIST recurrence from NIH also include rupture and location, but the mitotic index and size are the most important for any location. It wasn't mentioned in the manuscript.
Author Response
Major
The study concept was well prepared; however, this study doesn't show a clear way of relation with malignancy. In the manuscript, two definitions are used- recurrence and malignancy, which were not clearly defined. The reader can be confused, especially when in Table no 1, benign and malignant lesions were presented.
Also, there is no visible what patients were included to follow up in the method. Therefore, I propose to change and organize the method part, by adding primary and secondary outcomes.
☞ Thank you for your good comment. The primary outcome we wanted to present was malignancy, and the secondary outcome was recurrence. As you pointed out, it may be confusing because the comments and definitions for this part are not clear. Therefore, we presented the method for primary and secondary outcomes from page 4, line 88 to page 4, line 90.
…. Primary outcome was defined as malignant change, and secondary outcome was defined as recurrence. Malignant changes at the time of diagnosis, duration of follow - up and recurrence during follow-up were evaluated.
Minor
Based on the manuscript, the main tool, like the GIST volume, was not presented as a standardized measurement method? This main tool could be wider presented in the method/material part.
☞ We can assess the GIST volume through three dimensional computed tomography (CT). We presented the main tool in the method/material part from page 4, line 84 to from page 4, line 85.
… Through the cooperation of the radiology department, the patient's CT was reconstructed into 3D, and the radiology specialist evaluated the 3D volume through the reconstructed CT.
The recommendations for GIST recurrence from NIH also include rupture and location, but the mitotic index and size are the most important for any location. It wasn't mentioned in the manuscript.
☞ As you pointed out, we didn’t mention about rupture and location in the manuscript. We added related content in the introduction part from page 3, line 56 to page 3, line 58.
… They proposed anatomic site, size, mitotic rate, rupture, and location as conventional risk factors of GIST and divided recurrence potential into four risk groups by according to presence of these risk factors.
Round 2
Reviewer 2 Report
Dear authors,
Thank you for your changes.
I would also ask if you assessed the volume as a better indicator than size, can this method be applicable for the standard centers or only experts? If yes, please add it as an advantage.
The methods and material can not be accepted in the presented form. Please make an order in this section and create paragraphs eg study design, outcomes, statistic..
Also, the Results in the Abstract have to be improved and contain a crucial part of the results.
As you highlighted the primary (malignancy) and the secondary (recurrence) outcomes, please include it in the abstract and conclusion as the important points of this study.
Instead of using the name of countries in the Discussion, please use the names of authors...
52 would prefer to use pathologies instead of diseases
Author Response
I would also ask if you assessed the volume as a better indicator than size, can this method be applicable for the standard centers or only experts? If yes, please add it as an advantage.
☞ Thank you for your good comment. Three-dimensional reconstruction is possible in the standard center, and volume measurement after three-dimensional reconstruction can be performed by radiologists, not just experts in related fields, so it can be presented as an advantage. Related parts were mentioned from page 8, line 194 to page 8, line 196
… This study has shown that volume is an effective measure of the risk of malignancy in patients with GIST. Three-dimensional reconstruction and volume measurement are possible in the standard center. Therefore, volume measurement at GIST can be used as a practically useful method for treat.
The methods and material can not be accepted in the presented form. Please make an order in this section and create paragraphs eg study design, outcomes, statistic..
Also, the Results in the Abstract have to be improved and contain a crucial part of the results.
As you highlighted the primary (malignancy) and the secondary (recurrence) outcomes, please include it in the abstract and conclusion as the important points of this study.
☞ The contents of the abstract and method part have been revised according to the points you suggested. The revised content is displayed in red letters and uploaded as a revised version.
Instead of using the name of countries in the Discussion, please use the names of authors...
☞ The contents of the discussion have been revised according to the points you suggested page 8, line 179 and page 8, line 181
…According a study published by Dematteo RP et al. in 2008, high recurrence rate is correlate with ≥ five mitoses per high power field, tumor length ≥ 10cm, and tumor location. [13] In a study conducted by Jumniensuk C, metastasis was more frequent in tumors ≥ 10 cm (p = 0.023), tumors of non-spindle cell type (p = 0.027), mitotic count ≥ 5 (p = 0.000), and myxoid change (p = 0.002) and recurrence was associated with myxoid change (p = 0.045).